# Molecular Investigation of Recent Canine Parvovirus-2 (CPV-2) in Italy Revealed Distinct Clustering

**DOI:** 10.3390/v14050917

**Published:** 2022-04-28

**Authors:** Marilena Carrino, Luca Tassoni, Mery Campalto, Lara Cavicchio, Monica Mion, Michela Corrò, Alda Natale, Maria Serena Beato

**Affiliations:** 1Diagnostic Virology Laboratory, Istituto Zooprofilattico Sperimentale delle Venezie (IZSVe), Viale dell’Università 10, 35020 Legnaro, Italy; mcampalto@izsvenezie.it (M.C.); lcavicchio@izsvenezie.it (L.C.); mmion@izsvenezie.it (M.M.); mcorro@izsvenezie.it (M.C.); anatale@izsvenezie.it (A.N.); 2Virology Laboratory, Istituto Zooprofilattico Sperimentale dell’Umbria e delle Marche “Togo Rosati”, Via G. Salvemini 1, 06126 Perugia, Italy; ms.beato@izsum.it

**Keywords:** CPV-2, *Carnivore Protoparvovirus 1*

## Abstract

Canine parvovirus Type 2 (CPV-2) is a worldwide distributed virus considered the major cause of viral gastroenteritis in dogs. Studies on Italian CPV-2 are restricted to viruses circulating until 2017. Only one study provided more updated information on CPV-2 but was limited to the Sicily region. No information regarding the circulation and genetic characteristics of CPV-2 in Northeast Italy has been made available since 2015. The present study investigated the genetic characteristics of CPV-2 circulating in the dog population of Northeast Italy between 2013 and 2019. The VP2 gene of 67 CPV-2 was sequenced, and phylogenetic analysis was performed to identify patterns of distribution. Phylogenetic and molecular analysis highlighted unique characteristics of Northeast Italian CPV-2 and interestingly depicted typical genetic clustering of the Italian CPV-2 strains, showing the existence of distinct CPV-2 genetic groups. Such analysis provided insights into the origin of some Italian CPV-2 genetic clusters, revealing potential introductions from East European countries and the spread of CPV-2 from South/Central to North Italy. This is the first report that describes the genetic characteristics of recent Italian CPV-2. Tracking the genetic characteristics of CPV-2 nationally and globally may have impact on understanding the evolution and distribution of CPV-2, in particular in light of the current humanitarian emergency involving Ukraine, with the massive and uncontrolled movement of people and pet animals.

## 1. Introduction

Canine parvovirus Type 2 (CPV-2) belongs to the *Parvoviridae* family, *Parvovirinae* subfamily, *Protoparvovirus* genus and *Carnivore Protoparvovirus 1* species. CPV-2 was firstly identified in 1977 as the causal agent of severe gastroenteritis among groups of dogs, previously ascribed mainly to colibacillosis infection or individual animal problems of noninfectious origin [1]. CPV-2 is transmitted through the fecal–oral route; it replicates in lymph nodes, spreads in the bloodstream and reaches the gastrointestinal tract. Following 3–7 days of incubation, characteristic signs of infections occur, including depression, loss of appetite, vomiting, high fever, severe diarrhea and myocarditis [2]. It is highly contagious in puppies up to six months [1], especially between four and twelve weeks of age, when maternally derived antibodies progressively decline, making a puppy susceptible to CPV-2 infection [3]. Vaccination at the age of six–eight weeks is the most effective preventive tool available [4], although the occurrence of novel CPV-2 variants might negatively impact the efficacy of vaccines currently available [5].

CPV-2 is a small, non-enveloped virus with a 26 nm diameter icosahedral capsid that holds a 5323 nucleotide (nt) single-stranded (ss) negative DNA genome [6,7]. The DNA molecule is composed of two Open Reading Frames (ORFs). The first ORF (ORF1), encoding non-structural proteins NS1 and NS2, is located at the 3′ end of the genome, and the second ORF (ORF2), located at the 5′ end, encodes three structural proteins: VP1, VP2 and VP3. The VP2 protein (64 kDa) is a truncated form of VP1 (84 kDa) deriving from alternative splicing of the same mRNA. The VP3 protein derives from the post-translational proteolytic cleavage of 15–20 amino acids (aa) in the amino terminus of VP2, and it is only present in DNA-containing virions [2,8]. Full capsids are made of 60 copies of VP2, VP3, and VP1 combinations, where VP2 is the most abundant capsid protein (90%) [2,8]. VP2, being the major capsid protein, represents the main determinant of host range and it is subjected to antibody-mediated selection [9]. Moreover, the VP2 protein presents a high variability and therefore is the major target of CPV-2 characterization and phylogenetic studies [10,11,12,13,14].

Regarding CPV-2 evolution, two different hypotheses exist. Early studies maintained that CPV-2 emerged as a host variant of Feline Panleukopenia virus (FPLV) or FPLV-like virus [15,16]. By contrast, recent studies reported that CPV-2 and FPLV viruses derive from a common ancestor of unknown origin and evolved independently [17]. Although CPV-2 is a DNA virus, it evolves rapidly, showing a very high genomic substitution rate similar to those of RNA viruses [18]. Following its emergence in 1977, CPV-2 underwent rapid evolution, and new antigenic variants arose worldwide [16,19,20,21,22,23,24]. Following the emergence of the CPV-2, identified as the original type (CPV-2 OR), several antigenic variants were identified and described as CPV-2a, 2b, 2c, new2a and new2b [16,19,20,21,22,23,24]. The CPV-2a strains present A300G substitution; CPV-2b and CPV-2c show an additional aa mutation: the N426D and N426E, respectively [16,19,20,21,22]. CPV-new2a and CPV-new2b variants are both characterized by an additional aa mutation compared to those of the CPV-2a and 2b variants: the S297A [23,24]. The circulation of CPV-2 variants has become predominant with a decrease in CPV-2 OR detection; however, it is still detected in dogs, probably due to its use as a vaccine seed strain [25]. Beyond the field variants and CPV-2 OR, other variants identified as vaccine-derived strains are also detected in dogs, namely the Intervet strain (CPV-2 INT) (contained in the Nobivac Puppy DC vaccine) [26] and the Pfizer strain (CPV-2 PF) (contained in the Vanguard vaccine) [27].

All CPV-2 variants have been reported worldwide and coexist in different ratios in dog populations [28,29]. Regarding the presence and distribution of CPV-2 variants in Italy, several studies have been conducted so far, and together cover a timeframe from 1994 to 2019 [9,11,12,30,31].

Only one study reports the presence and characterization of CPV-2 in Northeast Italy [11]. In this study, the authors described the genetic characteristics of 100 CPV-2 strains, collected in several Italian regions from 2008 and 2015, including 32 CPV-2 strains from Northeast Italy. Therefore, no updated information is available regarding the circulation of CPV-2 variants and their genetic characteristics in Northeast Italy from 2015 onwards. This area of Italy is of particular interest as a point of entry of pet animals from Eastern European countries, via legal or illegal movements [32,33], demonstrated by recent studies on the characterization of pathogens of dogs including canine parvoviruses from East Europe and Far East countries [32,33]. The aim of the present study was to characterize the VP2 of CPV-2 strains identified through the diagnostic activity detected between 2013 and 2019 in Northeast Italy, providing updated information on the circulation and distribution patterns of CPV-2 variants.

## 2. Materials and Methods

### 2.1. Samples Analyzed during the Routine Diagnostic Activity in Northeast Italy

Two hundred and ninety-three samples were included in the present study, collected between 2013 and 2015 and between 2017 and 2019 in Northeast Italy. Samples were received at the diagnostic virology laboratory of the Istituto Zooprofilattico Sperimentale delle Venezie (IZSVe) for parvovirus detection. Samples were submitted by private animal clinics or by peripheral IZSVe diagnostic laboratories, which received carcasses from necropsy collected from Northeast Italy (Veneto, Friuli Venezia Giulia regions and Trento and Bolzano autonomous provinces) and from other Italian regions (Emilia Romagna, Marche and Puglia) in Central and South Italy, in addition to few samples from Hungary. A descriptive analysis of collected data such as origin, age, sex, breed and vaccination status was performed, and the qualitative and quantitative variables are displayed in tables and graphs. Some carcasses were conferred for a specific suspected diagnosis. Others were collected by the Public Veterinary Services within passive surveillance activities. In this study, CPV-2 was investigated based on suspected pathological lesions observed during necropsy.

### 2.2. Detection and Characterization of CPV-2 in Dog Tissues

#### 2.2.1. Sample Preparation and DNA Extraction

A section of intestine samples (about 5 mm^3^) was homogenized in 0.8 mL of Phosphate-Buffered Saline (PBS) supplemented with antibiotics (PBS-A: 10,000 IU/mL penicillin G, 10 mg/mL streptomycin, 5000 IU/mL nystatin and 0.25 mg/mL gentamicin sulfate), using Tissue Lyser (QIAGEN, Hilden, Germany) at 30 Hz for 3 min. Two hundred microliters (µL) of intestine homogenates was used for the extraction of viral DNA using the “Nucleospin Tissue” (Macherey-Nagel, Düren, Germany) according to the manufacturer’s instructions. Homogenates and eluted DNA were stored at −80 °C until use.

#### 2.2.2. Real Time PCR

The eluted DNA was used for Real-Time PCR targeting a region of the VP2 gene of CPV-2 developed at the IZSVe. The following primer and probe sequences were used: CPV-2 for 5′-GATCCAATTGGAGGTAAAACAGG-3′, CPV-2 rev 5′-TTCTTTATCCCAAATTTGACC-3′ and CPV-2 probe FAM 5′-TGGTCCTTTAACTGCATTAAATAATGTACC-3′ TAMRA. Real-Time PCR was carried out in 25 µL reaction volume consisting of 5 µL of Quantifast Pathogen Master Mix [5×] (QIAGEN, Hilden, Germany), 1.5 µL of each primer (10 µM), 0.5 µL of probe (10 µM), 2.5 µL of Internal Control Assay (10×) (QIAGEN, Hilden, Germany) and 9 µL of sterile ultra-pure water. The PCR program used had the following thermal profile: Taq polymerase activation at 95 °C for 5 min, followed by 40 cycles with denaturation at 95 °C for 15 s and annealing at 59 °C for 30 s. Real-Time PCR was performed using a Biorad CFX96 instrument (Biorad, Hercules, CA, USA).

### 2.3. VP2 Sanger Sequencing

The VP2 gene of all CPV-2-positive samples detected between 2017 and 2019 and a selection of older samples collected between 2013 and 2015 were subjected to Sanger sequencing. Three different end-point PCRs were performed using a combination of three different primer pairs (2161F 5′-TTGGCGTTACTCACAAAGACGTGC-3′ and R2 5′-AAACTTTAGTTGGTGGCTGAG-3′; 1270F 5′-TGGAAATCACAGCAAACTC-3′ and 1270R 5′-AGTCTTGGTTTTAAGTCAGTATC-3′; F4 5′-GCTACCAACAGATCCAATTG-3′ and 4823R 5′-ACCAACCACCCACACCATAACAAC-3′) previously published [11,34,35], all covering the VP2 gene. The end-point PCRs were conducted with 5 µL of PrimeSTAR GXL Buffer (5×) (Takara, Kusatsu, Japan), 1.25 µL of each primer (10 µM), 2 µL of dNTPs (2.5 mM), 0.5 µL PrimeSTAR GXL Taq (1.25 U/µL) (Takara, Kusatsu, Japan), 10 µL of sterile ultrapure water and 5 µL of sample DNA to a total volume of 25 µL. The following thermal cycling profile was performed using a Biorad 1000S Thermal Cycler instrument (Biorad, Hercules, CA, USA): 94 °C for 5 min; followed by 35 cycles of 94 °C for 30 s, 54 °C (for 2161F-R2 and F4-4823R primer pairs) or 55 °C (for 1270F-1270R primer pair) for 30 s and 72 °C for 30 s; and a final extension at 72 °C for 8 min. Amplified DNA was subjected to electrophoresis in 7% acrylamide gels at 200 V for 45 min and analyzed after silver staining. Sanger sequencing of all positive samples was performed with the Ab3130xl instrument (ThermoFisher, Carlsbad, CA, USA) using the same primer pair used for amplification. Seqscape v2.6 was used to elaborate the raw data, and the obtained sequences were compared with those available in Genbank using BLAST [36]. Sequence data generated in the present study were submitted to the DDBJ/EMBL/GenBank databases under accession numbers from MT996017 to MT996061, MZ476774 and from OM525664 to OM525684 and are reported here as IZSVe CPV-2 strains or sequences.

### 2.4. Phylogenetic Analysis of the VP2 Gene

A first preliminary phylogenetic analysis was performed to identify the main genetic groups to which IZSVe CPV-2 sequenced samples belonged. For this purpose, an extended dataset was created starting from IZSVe CPV-2 sequences and including the first 100 blast results for a total of 1298 sequences [36]. Feline Panleukopenia virus (FPLV) sequences (n = 7) were used as an outgroup. All the duplicate sequences were removed, and the preliminary analysis, including 1327 sequences, was performed using the Maximum Likelihood (ML) method implemented in Iqtree v1.6 [37] with 1000 replicates of ultrafast bootstrap (UFboot). The phylogenetic tree was visualized with Figtree v.1.4.3 (FigTree v1.4.3, n.d.). Based on UFboot values and topology, genetic groups composed of sequences of the same antigenic variant were identified and named using the current nomenclature system (i.e., 2a, 2b, 2c) [16,20,21] followed by a progressive number to discriminate between them. From this first descriptive phylogenetic tree, a further analysis was conducted using a reduced dataset representative of the topological diversity of the preliminary tree that includes IZSVe CPV-2 sequences, the five most related sequences retrieved in GenBank and the sequences belonging to the genetic group identified in the preliminary analysis. When multiple identical sequences with the same epidemiological information (i.e., same location, country and year) were available, only one of them was maintained in the dataset. A total of 527 sequences were included in the phylogenetic analysis performed with Iqtree v1.6 with 100 non-parametric bootstrap replicates. The genetic groups identified in the preliminary analysis were colored, and subgroups were collapsed to facilitate the visualization. Only bootstrap values above 60 were displayed, and the final figure was edited with Inkscape v.1.0.2.

### 2.5. Identification of Unique Nucleotide and Amino Acid Mutations in Recent Italian CPV-2 Strains

We aimed to identify the presence of unique nt mutations in the VP2 gene of the IZSVe CPV-2 strains identified during 2013–2015 and 2017–2019, compared to all the available Italian sequences. All Italian CPV-2 sequences (n = 568) publicly available were downloaded from NCBI. The database was last accessed on 26th December 2021 and only included CPV-2 sequences whose origin was indicated as “Italy” in the sample Genbank record page. The nt sequences were then analyzed to identify unique nt and aa mutations and further compared with all Italian CPV-2 sequences available in Genbank.

## 3. Results and Discussion

All (n = 293) samples submitted for parvovirus detection to the diagnostic virology laboratory at IZSVe between 2013 and 2015 and between 2017 and 2019 were intestinal. In total, 161 out of 293 (54.95%) intestine samples were positive for CPV-2 infection, of which, 21 were collected during 2013, 55 during 2014, 37 during 2015, 11 during 2017, 29 during 2018 and 8 during 2019 (Figure 1).

VP2 sequencing was attempted for a selection of positive samples from 2013 to 2015 (n = 21) and for all (n = 48) positive samples from 2017 to 2019. The anamnestic data of 69 (21 + 48) samples out of 161 CPV-2-positive samples were available and are presented in Table 1.

Thirty-four dogs presented intestinal lesions, seven did not present any intestinal lesions, and for twenty-eight dogs, this information was not available (Table 1). The vaccination history was only known for 2 out 69 CPV-2-positive samples (MT996036 CPV2-Canis-Italy-Veneto-strain 17DIAPD-52386-3-2017-new2b and MT996046 CPV2-Canis-Italy-Veneto-strain 18DIAPD-54123-2-2018-2c) (Table 1). Forty-three dogs were puppies (2.5 months of median age, twenty-six males and twelve females; for five samples, the sex information was not available (N/A)); fourteen dogs were adults (median age 2 years, six males, seven females and one not available); for eighteen dogs, the age information was not available (Table 1). Samples were collected from nine mixed-breed dogs and forty different recognized breeds, and for two dogs, the breed information was absent (Table 1). Sanger sequencing of the VP2 gene was attempted for 69 samples, and a total of 67 novel CPV-2 VP2 sequences were successfully generated. Two positive samples collected in 2018 were not successfully sequenced and typed. The geographic origin of sequenced CPV-2 is reported in Figure 1 and Figure 2.

Of the 67 sequences, eight were from 2013, seven were from 2014, six were from 2015, eleven were from 2017, 27 were from 2018 and eight were from 2019 (Figure 1). For 66 out of 67 sequences, the CPV-2 variant was inferred from sequence analysis, and one out 67 sequences was untypable due to the presence of a degenerate nucleotide in the nucleotide residues 1276-1278, which are responsible for CPV-2 classification (Figure 2). The sequencing of 67 CPV-2 strains detected in the present study and identified as IZSVe CPV-2 strains or sequences yielded 28 full-length VP2 and 39 partial-length VP2 sequences. Three out of five field CPV-2 variants described so far were detected in the present study, in addition to the CPV-2 OR and namely: new2a, new2b and 2c. The 2a and 2b variants were not detected, differently from previous reports where CPV-2a and 2b were identified and the 2a variant was described as the most prevalent variant circulating in Italy from 1994 to 2017 [11,12]. The present study identified the new2a as the most prevalent variant identified between 2013 and 2019, followed by new2b and 2c. In detail, the new2a variant was identified in 34 samples (50.75%); 5 in 2013 (of which 1 was of Hungarian origin), 5 in 2014 (of which 1 was of Hungarian origin), 2 in 2015, 3 in 2017, 17 in 2018 and 2 in 2019; the new2b in 12 (17.91%) samples, 1 in 2012, 1 in 2013, 2 in 2017 and 8 in 2018; and the 2c variant in 13 samples (19.40%), 1 in 2012, 1 in 2013 of Hungarian origin, 4 in 2015, 3 in 2017, 1 in 2018 and 3 in 2019 (Figure 1 and Figure 2). The CPV-2 OR was identified in two (2.99%) samples (one in 2017 and one in 2018). In addition, five samples were characterized as vaccine strains: four (5.97%) samples were identified as the Intervet Vaccine (INT) (one in 2017, one in 2018 and two in 2019) and one (1.49%) sample as the Pfizer Vaccine (PF), collected in 2017 (Figure 1 and Figure 2). Of note, all samples characterized as CPV-2a and 2b, in the present study, were of the subtype “new”, therefore presenting the aa mutation A297S, suggesting that the old variants of CPV-2a and CPV-2b might have been replaced by new variants in Northeast Italy.

The prevalence of the CPV-new2a variant is a new piece of information regarding the Italian peninsula; in fact, previous studies reported a higher prevalence of CPV-2a in Italy until 2015 [11,12]. Nevertheless, a different scenario was described in Italian islands such as Sicily [30] and Sardinia [31], where CPV-2c and CPV-new2b were the most prevalent variants, respectively, between 2009 and 2019 and 2005 and 2013. Data presented herein show that the CPV-new2a is also the most commonly detected variant at a regional level during the study period (Figure 2). However, such an observation needs additional confirmatory data to establish whether the proportions in variants reflect differences in prevalence. The phylogenetic analysis, based on the VP2 nt sequence of 527 CPV-2 strains, highlighted that the phenotypic separation among variants was partially reflected in the tree topology, and a certain geographical clustering of Italian strains was visible. Here, we describe eight monophyletic groups containing the majority of the IZSVe CPV-2 strains and here identified as: OR, INT, 2A.1, 2A.2, 2A.3, 2B 2C.1 and 2C.2 (Figure 3A–C and Appendix A).

Forty-nine IZSVe CPV-2 strains out of sixty-seven (73.13%) fell into one of the eight newly described genetic groups, together with other exclusively Italian strains (2B group) or with the majority of Italian strains, including foreign ones (OR, INT, 2A.1, 2A.2, 2A.3, 2C.1 and 2C.2 groups) (Figure 3A–C and Appendix A). The remaining IZSVe CPV-2 strains (n = 18) appeared scattered in the phylogenetic tree (Figure 3A–C and Appendix A), with 17 out 18 belonging to the CPV-new2a variant and 1 out 18 being untypable. Interestingly, three genetic groups were composed mainly (2C.1, 2A.1 and 2A.3) or exclusively (2B) of Italian CPV-2 strains. The phylogenetic analysis showed that three genetic groups indicated as 2A.1, 2A.2 and 2A.3 (Figure 3A and Appendix A) contained exclusively CPV-new2a strains, of which, seventeen were identified in the present study. The 2A.1 group (Figure 3A and Appendix A) included Italian CPV-new2a strains detected between 2008 and 2018, with one sequence from the UK (collected in 2017), one from Portugal (collected in 2014) and one from Hungary (collected in 2013). The 2A.1 group strains and related ones, therefore, appear to have circulated in Italy since 2008 with the incursion of similar strains from foreign countries and circulation in wild animals, such as in otters (MG977495 and MG977499) (Figure 3A and Appendix A). As a matter of fact, the oldest sequence of 2A.1 group was identified in 2008 from the Lombardia region in North Italy (MH491849) that likely spread in northern Italy in the following years (Figure 3A and Appendix A). The 2A.2 group contained three CPV-new2a strains characterized in the present study, and all were identified in 2018 in Northeast Italy (two from the Friuli Venezia Giulia region and one from Veneto), and three CPV-new2a sequences from China all detected between 2013 and 2014, with one sequence for which the year of detection was not available (Figure 3A and Appendix A). This group was characterized by the nt mutation T1485C (Table 2), described for the first time in Italy and interestingly possessed by all the three Chinese strains present in this group.

The third group containing IZSVe CPV-new2a strains was 2A.3, characterized by eleven Italian CPV-2 strains detected between 2013 and 2018 (four out of eleven identified in the present study) and interestingly, by foreign sequences exclusively from Hungary all (n = 4) detected in 2012 (Figure 3A and Appendix A). In detail: two IZSVe CPV-2 strains were identified in 2018 from Trentino Alto Adige, clustering with an Italian CPV-2 strain identified from a wolf in Central Italy (Abruzzo-MT454914), one IZSVe CPV-2 from Friuli Venezia Giulia in 2017 and one from Veneto in 2018. The remaining six Italian CPV-2 sequences of this group were identified, with four from Veneto from 2014 and 2018 and two from Sardinia identified in 2013 (Figure 3A and Appendix A). The two IZSVe CPV-2 strains from Trentino Alto Adige identified in 2018 shared an nt mutation in the VP2 gene: the A156G (Table 2). This small subgroup contains a sequence from a wolf (Abruzzo-MT454914) for which it was not possible to assess the presence of the A156G mutation due to lack of sequencing coverage. The twelve CPV-new2b strains identified in the present study were all grouped in the group 2B, with other exclusively Italian CPV-2 strains (Figure 3B and Appendix A). In detail, this cluster contained CPV-new2b Italian strains from Northeast Italy (n = 11, 10 out of 11 identified in the present study) detected between 2015 and 2018, a strain from North West Italy (n = 1) detected in 2012, strains from Central Italy (n = 10, 1 out of 10 identified in the present study) detected between 2008 and 2018, and strains from South/insular Italy (n = 4, 1 out of 4 identified in the present study) detected between 2011 and 2018 (Figure 3B and Appendix A). The remaining ten strains of this group were detected between 2009 and 2017, for which the region of origin is not available (Figure 3B and Appendix A). However, the oldest dog CPV-new2b strain of this group was detected in Central Italy in 2008 (MH491853 5733_Italy_Lazio_14_06_2008) and likely spread to North and South Italy. Interestingly, the 2B group is characterized by a small sub-group composed of three IZSVe CPV-2 sequences (MT996025, MT996020 and MT996036) that all present the mutation T459C (Table 2). The 2C groups were formed by CPV-2c strains displaying broader heterogeneity, as various geographic origin strains fall into 2C.1 and 2C.2 groups and are specifically from Europe, Asia and Africa (Figure 3C and Appendix A). The IZSVe CPV-2c strains appear scattered within the two 2C groups. Eight IZSVe CPV-2c strains fall into the 2C.1 group and five into the 2C.2 group (Figure 3C and Appendix A). The IZSVe 2C.1 strains were detected from 2015 to 2019, and the IZSVe 2C.2 strains were detected from 2013 to 2019 (Figure 3C and Appendix A). The phylogenetic analysis showed a distinct clustering of the vaccine CPV-2 strains divided into two main groups: CPV-2 OR and CPV-2 INT (Figure 3B and Appendix A), where two IZSVe CPV-2 OR and four IZSVe CPV-2 INT strains, respectively, fall into these groups (Figure 3B and Appendix A).

The presence of unique aa mutations was investigated in the IZSVe CPV-2 strains under study. Multiple alignment of IZSVe CPV-2 nt sequences and all (n = 568) Italian CPV-2 revealed the presence of several point-mutations observed exclusively in the VP2 gene of recently identified CPV-2 strains in Northeast Italy (Table 2). The presence of 26 nt point-mutations was observed: 12 mutations were silent and 14 were missense, all described here for the first time in the Italian scenario (Table 2).

Unique aa mutations were identified in ten IZSVe CPV-2 strains: three strains belonged to the new2a variant, one falling in group 2A.1, one in group 2A.3 and one did not belong to any of the newly identified monophyletic group; three to the new2b variant, two to variant 2c and two to CPV-2 OR (Table 2). Ten out of fourteen unique aa mutations fell between position 1024 and 1297 (Table 2). The IZSVe CPV-2 OR strains both presented two unique aa mutations compared with other Italian sequences: K271R and V316I (Table 2). It should be noted that some non-Italian CPV-2 strains clustering in the CPV-2 OR group possessed the K271R and V316I mutations, suggesting that they might indicate a CPV-2 OR introduction from abroad (data not shown). The CPV2-Canis-Italy-Veneto-15VIR1684-6-2015-2c presented two mutations: P352S and F353I, and the CPV2-Canis-Italy-Veneto-13VIR2697-8-2013-new2b presented four unique aa mutations: V308G; A347S; S348F and G360E (Table 2). Interestingly, all but one of the unique aa mutations described here for the first time in Italy are possessed by CPV-2 strains circulating in Northeast Italy.

CPV-2 emerged at the end of the 1970s and rapidly spread, reaching global circulation. Following 40 years of circulation, limited genetic diversity was observed with the accumulation of few fixed mutations [38]. Despite the worldwide distribution of CPV-2 and the low genetic diversity, unexpected clustering was observed for the Italian CPV-2 strains. The acquisition of a relevant number of CPV-2 sequences from Northeast Italy generated in this study provided an updated and detailed picture of the viral distribution patterns in Italy. The phylogeny of CPV-2 strains from Northeast Italy from 2013 to 2019 highlighted peculiar clustering of Italian CPV-2 strains in general. The differentiation of the CPV-2 variants circulating in Italy appears to be only partially reflected in the tree topology. This pattern was already described by Voorhees and colleagues [38], who highlighted how the aa changes that determine the antigenic variants do not correspond to supported monophyletic groups. They also suggested that these changes are likely to have arisen multiple times after the emergence of CPV-2a, and this could explain why antigenic variants do not constitute single monophyletic groups, as observed in the present study. However, we could identify eight interspersed monophyletic groups, each composed of a single antigenic variant, in which most IZSVe CPV-2 sequences could be grouped. The clustering of CPV-2 characterized in the present study with previously deposited Italian sequences suggests the free circulation, without regional barriers, of Italian CPV-2 within the country as previously noted [11]. This is evidenced by the coexistence of the same CPV-2 variants in different Italian regions and by the presence of CPV-2 from different Italian regions in the same monophyletic group. Therefore, frequent transmission events can be hypothesized across Italian regions sustained by the intense movement of persons and their pets. Regarding the groups observed and described for the first time here, the absence of strains identified before in 2018 in the 2A.2 group poses questions on the origin of these strains clustering with Chinese CPV-2 sequences. Such evidence might be explained by the lack of proper surveillance in Italy, and therefore, the lack of detection of CPV-2 strains that may fall into this group or the introduction of CPV-2 strains from foreign countries, in particular from the Far East. As a matter of fact, the CPV-2 strains composing this group were identified in puppies suspected to be illegally imported into Italy (data not shown). The introduction of CPV-2-positive puppies in Northeast Italy between 2017 and 2020 was documented by an intense interregional surveillance program between Italy and Austria on the illegal movement of pet animals, as Northeast Italy represents an easy point of entry for Eastern European countries [32]. Therefore, the hypothesis on the introduction of genetically different CPV-2 strains into Italy from foreign countries cannot be ruled out. This is also supported by the presence of Hungarian strains clustering into the 2A.3 group. The genetic picture for CPV-2 in Italy appears to be dictated by both legal and illegal movements of companion animals with human migration and travel at the national and international levels, affecting the spread and evolution of CPV-2. This highlights that the continuous national and international monitoring of CPV-2 may aid the understanding of CPV-2 distribution and the tracking of the introduction of strains from foreign countries, as previously demonstrated for other viruses affecting the dog population [33]. CPV-2 infection presents some peculiarities such as the endemicity status and the vast use of live attenuated vaccines formulated with different CPV-2 variants that render fundamental matching of the collection and analysis of clinical/anamnestic data with virological ones.

A major limitation to the study of CPV-2 distribution and characterization is the absence of anamnestic data that makes it difficult to provide any hypothesis on the origin of the CPV-2 variants in a specific geographic area. This applies in particular to the CPV-2 OR strains considered to have disappeared worldwide [16,20,25], probably due to the use of vaccines based on this variant, but indeed, few cases have still been detected. The detection of the CPV-2 OR variant therefore could be explained by the re-isolation of the vaccine strain or the circulation of CPV-2 OR field variants [25]. Coupling the molecular analysis of VP2 and NS1/NS2 [9] with the collection of detailed clinical and anamnestic data regarding the vaccination status of dogs presenting gastroenteritis signs, the vaccine used and the vaccination protocol applied, may elucidate the origin and therefore the meaning of the detection of the vaccine and other variant strains.

The fourteen aa mutations reported herein for the first time for CPV-new2a, new2b and 2c should be further investigated in order to better understand whether they may continue to appear in other Italian strains that circulate in Northeast Italy or/and in other regions and whether they might represent signatures of introduction from abroad. In addition to aa mutations dictating the antigenicity of strains and differentiating them in CPV-2a, CPV-2b and CPV-2c, several studies described mutations in various VP2 residues. Substitution I418T was detected in CPV-2a and CPV-2b strains identified in Italy [21,31,39,40] and South Korea [41], A371G was identified in Italian CPV-2b sequences [31] and in the Italian sequence KF373599 (for which there is no literature reference), T440A was reported in Italy [39], Korea [41], Uruguay [42], China [43], India [44] and Thailand [45] and T440P and Y324L were characterized exclusively in Italy (T440P: [31] and Y324L: [10]). Although CPV-2 displays a relatively low genetic diversity and accumulated few changes after its adaptation to the dog host [38], genetic surveillance to trace the appearance of novel mutations, especially in antigenic proteins such as VP2 protein, is of remarkable importance, since they might be responsible for vaccination failure.

## 4. Conclusions

Our study provides novel information regarding the genetic characteristics of recent Northeast Italian CPV-2, bringing to light peculiar genetic clustering of CPV-2 in Italy. For the first time, the present report describes the presence of new groups circulating in Italy. The phylogenetic analysis suggested the existence of genetic flows across Italy and a possible introduction of CPV-2 from East European and Asian countries. Diagnosis, the collection of anamnestic data, and sequencing of CPV-2 are all crucial steps for proper monitoring of this disease. Therefore, continuous efforts should be implemented in monitoring the CPV-2 in Italy to better understand the circulation pattern of its variants and the distribution of the genetic groups identified.

## Figures and Tables

**Figure 1 viruses-14-00917-f001:**
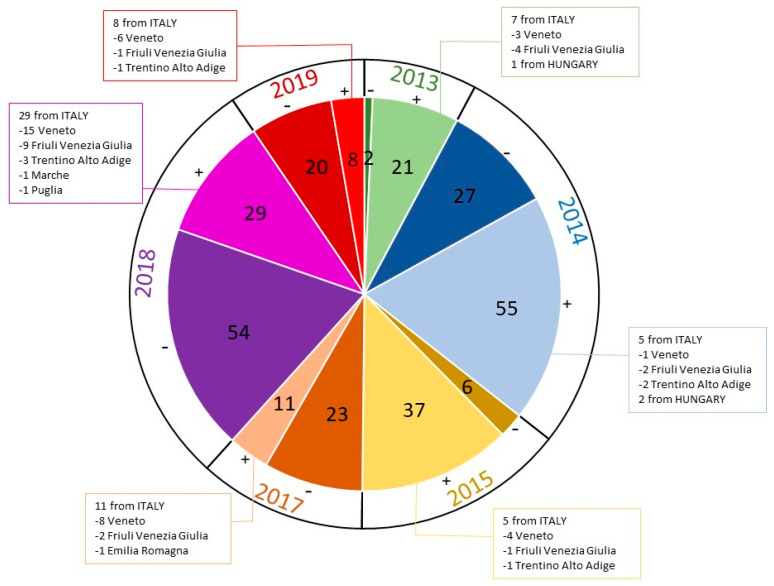
Pie chart of intestine samples analyzed for CPV-2 infection during 2013–2015 and 2017–2019. The pie chart reports the total number of intestine samples analyzed which were positive or negative for CPV-2 infection during 2013–2015 and 2017–2019. The lateral boxes report the state and region of origin of CPV-2-positive samples sequenced for phylogenetic analysis.

**Figure 2 viruses-14-00917-f002:**
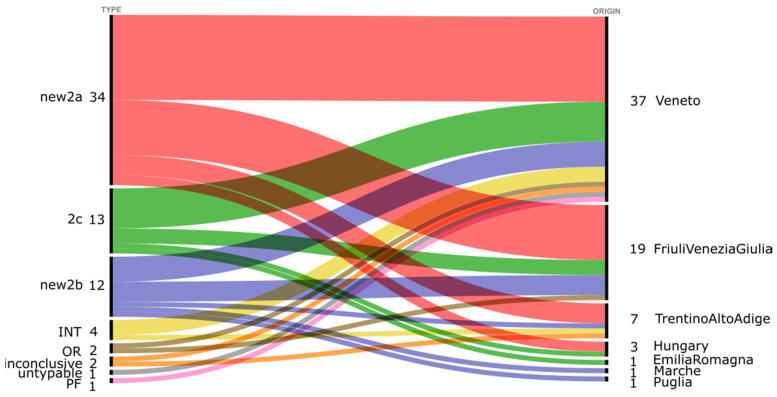
Alluvial plot of CPV-2 variants divided by region of origin. The left side of the diagram shows the CPV-2 variants identified in the present study: CPV-new2a (new2a-red), CPV-new2b (new2b-blue) and CPV-2c (2c-green), CPV-2 INT (Intervet Vaccine-yellow), CPV-2 OR (OR-brown) and CPV-2 PF (PF-purple). Inconclusive (orange): no consensus sequence obtained; untypable (grey): reported as untypable. The right side shows the geographic origin of CPV-2-positive samples. The width of the colored bands linking CPV-2 strains and region of origin is proportional to the number of isolates of each CPV-2 strain from each region of origin.

**Figure 3 viruses-14-00917-f003:**
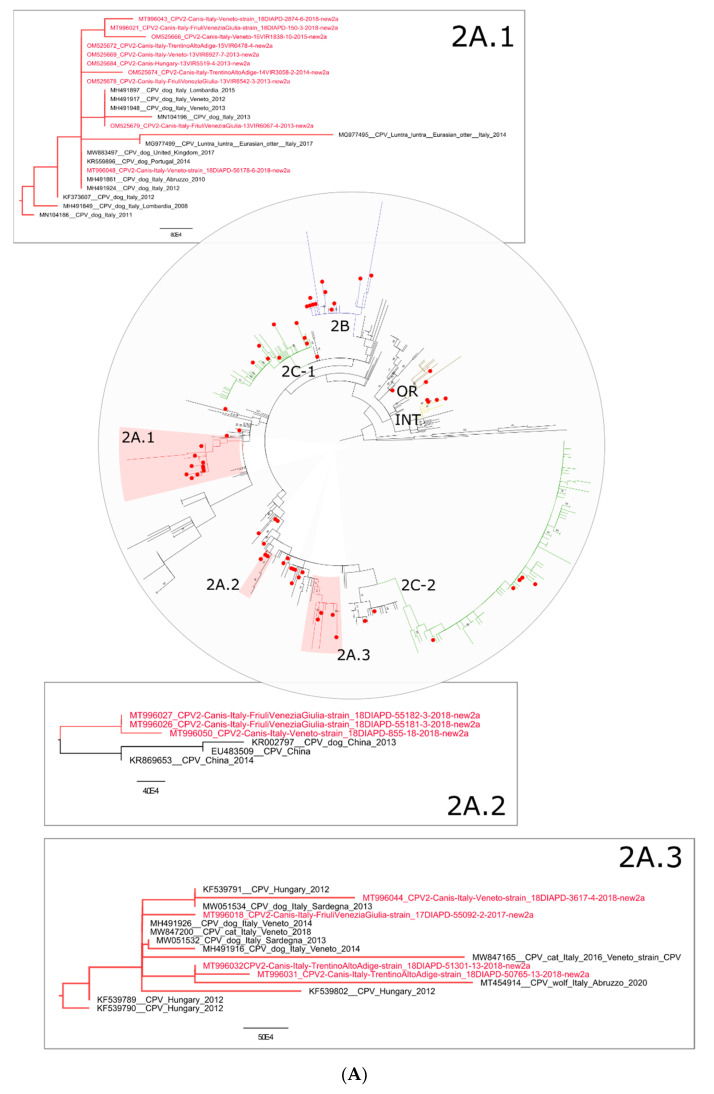
(**A**) Circular phylogenetic tree of the VP2 nucleotide sequences of CPV-2 identified in the present study (identified with a red dot) and similar CPV-2 strains available in GenBank. Groups 2A.1, 2A.2 and 2A.3 were highlighted in the circular tree and magnified in rectangular boxes. The IZSVe CPV-2 strains are highlighted in red. (**B**) Circular phylogenetic tree of the VP2 nucleotide sequences of CPV-2 identified in the present study (identified with a red dot) and similar CPV-2 strains available in GenBank. Groups 2B, OR and INT were highlighted in the circular tree and magnified in rectangular boxes. The IZSVe CPV-2 strains are highlighted in red. (**C**) Circular phylogenetic tree of the VP2 nucleotide sequences of CPV-2 identified in the present study (identified with a red dot) and similar CPV-2 strains available in GenBank. Groups 2C.2 and 2C.1 were highlighted in the circular tree and magnified in rectangular boxes. The IZSVe CPV-2 strains are highlighted in red.

**Table 1 viruses-14-00917-t001:** Information of CPV-2-positive samples included in the present study. Year of isolation, Italian region of origin and information regarding dog samples are reported. M = male, F = female, P = puppy, A = adult, Y = yes, N = no, N/A= not available information, d = days, m = months, y = years.

Sample	Year	Origin of the Sample	Breed	M/F	Puppy/Adult	Age	Intestinal Lesions	Vaccination
VIR-2391/6	2013	Veneto	Chihuahua	M	P	5 m	Y	N/A
VIR-2697/8	2013	Veneto	Rottweiler	F	A	6 m	N	N/A
VIR-5519/4	2013	Hungary	Yorkshire Terrier	M	P	1 m	N/A	N/A
VIR-5997/4	2013	Friuli Venezia Giulia	N/A	M	P	N/A	N	N/A
VIR-6066/4	2013	Friuli Venezia Giulia	N/A	M	P	N/A	N	N/A
VIR-6067/4	2013	Friuli Venezia Giulia	Dachshund	M	P	N/A	Y	N/A
VIR-6542/3	2013	Friuli Venezia Giulia	N/A	M	P	N/A	Y	N/A
VIR-6927/7	2013	Veneto	Maltese	F	P	1 m	Y	N/A
VIR-1263/3	2014	Hungary	Pincher	M	P	4 m	Y	N/A
VIR-3058/2	2014	Trentino Alto Adige	N/A	F	P	3 m	Y	N/A
VIR-3493/3	2014	Trentino Alto Adige	N/A	N/A	N/A	N/A	Y	N/A
VIR-5175/9	2014	Hungary	Siberian Husky	F	P	2 m	N	N/A
VIR-6046/5	2014	Friuli Venezia Giulia	Chow-Chow	M	P	4 m	Y	N/A
VIR-6666/2	2014	Friuli Venezia Giulia	N/A	N/A	N/A	N/A	N/A	N/A
VIR-7758/9	2014	Veneto	Yorkshire Terrier	F	P	2 m	Y	N/A
VIR-1684/6	2015	Veneto	N/A	F	N/A	N/A	Y	N/A
VIR-1838/3	2015	Veneto	N/A	F	N/A	N/A	Y	N/A
VIR-2037/1	2015	Veneto	N/A	N/A	N/A	N/A	N/A	N/A
VIR-6238/10	2015	Veneto	Mix breed	N/A	P	3.5 m	Y	N/A
VIR-6378/5	2015	Friuli Venezia Giulia	Maltese	M	P	2 m	Y	N/A
VIR-6478/4	2015	Trentino Alto Adige	N/A	N/A	N/A	N/A	N/A	N/A
DIAPD-1387/7	2017	Veneto	American Staffordshire Terrier	N/A	P	3 m	Y	N/A
DIAPD-1751/2	2017	Emilia Romagna	Labrador	M	P	50 d	Y	N/A
DIAPD-2046/16	2017	Veneto	Golden Retriever	N/A	P	4 m	Y	N/A
DIAPD-53054/6	2017	Veneto	Mix breed	M	A	2 y	Y	N/A
DIAPD-53504/6	2017	Veneto	Mix breed	M	P	65 d	Y	N/A
DIAPD-55092/2	2017	Friuli Venezia Giulia	N/A	F	N/A	N/A	N/A	N/A
DIAPD-56192/2	2017	Veneto	Siberian Husky	F	A	N/A	N/A	N/A
DIAPD-56643/4	2017	Veneto	Beagle	N/A	P	3 m	N	N/A
DIAPD-57033/8	2017	Veneto	Miniature Poodle	M	A	8 m	N/A	N/A
DIAPD-57861/2	2017	Friuli Venezia Giulia	English Pointer	F	A	5.5 m	N/A	N/A
DIAPD-766/5	2018	Veneto	Chihuahua	F	A	9 m	Y	N/A
DIAPD-1100/3	2018	Friuli Venezia Giulia	Mix breed	M	P	4 m	N/A	N/A
DIAPD-1399/7	2018	Veneto	Maltese	M	P	3.5 m	N/A	N/A
DIAPD-150/3	2018	Friuli Venezia Giulia	Poodle	M	P	2 m	Y	N/A
DIAPD-2874/6	2018	Veneto	Pincher	M	A	4 y	Y	N/A
DIAPD-3417/6	2018	Puglia	Mix breed	F	P	3 m	Y	N/A
DIAPD-3617/4	2018	Veneto	Mix breed	F	P	3 m	Y	N/A
DIAPD-50765/13	2018	Trentino Alto Adige	Shiba Inu	M	P	3 m	N/A	N/A
DIAPD-51020/3	2018	Friuli Venezia Giulia	Mix breed	F	P	2 m	N/A	N/A
DIAPD-51301/13	2018	Trentino Alto Adige	N/A	N/A	N/A	N/A	N/A	N/A
DIAPD-52097/6	2018	Marche	N/A	N/A	N/A	N/A	N	N/A
DIAPD-52386/3	2018	Veneto	American Staffordshire Terrier	M	A	3 y	Y	Y *
DIAPD-52568/4	2018	Veneto	N/A	F	A	4 y	N/A	N/A
DIAPD-52707/2	2018	Friuli Venezia Giulia	Mix breed	M	P	3 m	Y	N/A
DIAPD-52984/2	2018	Friuli Venezia Giulia	Mix breed	F	P	5 m	N/A	N/A
DIAPD-53807/2	2018	Friuli Venezia Giulia	N/A	F	A	8 m	Y	N/A
DIAPD-54123/2	2018	Veneto	N/A	N/A	A	2 y	N/A	Y *
DIAPD-55181/3	2018	Friuli Venezia Giulia	Italian Hound	M	P	3 m	N/A	N/A
DIAPD-55182/3	2018	Friuli Venezia Giulia	Italian Hound	M	P	3 m	N/A	N/A
DIAPD-55509/2	2018	Veneto	N/A	F	P	3 m	N/A	N/A
DIAPD-56178/6	2018	Veneto	N/A	N/A	N/A	N/A	N/A	N/A
DIAPD-56237/5	2018	Veneto	Poodle Toy	F	P	2 m	N/A	N/A
DIAPD-56531/4	2018	Friuli Venezia Giulia	Golden Retriever	M	P	2 m	N/A	N/A
DIAPD-56557/6	2018	Veneto	Alaskan Malamute	F	A	9 m	N/A	N/A
DIAPD-58450/8	2018	Trentino Alto Adige	N/A	N/A	N/A	N/A	N/A	N/A
DIAPD-855/18	2018	Veneto	Chihuahua	M	P	2 m	Y	N/A
DIAPD-855/19	2018	Veneto	Chihuahua	F	P	2 m	Y	N/A
DIAPD-855/42	2018	Veneto	Chihuahua	M	P	2 m	Y	N/A
DIAPD-855/6	2018	Veneto	Chihuahua	M	P	2 m	Y	N/A
DIAPD-883/6	2018	Veneto	Maltese	F	N/A	N/A	N/A	N/A
DIAPD-200/5	2019	Friuli Venezia Giulia	Boxer	M	A	6 m	Y	N/A
DIAPD-2836/7	2019	Veneto	Boxer	M	A	N/A	Y	N/A
DIAPD-3366/4	2019	Veneto	Yorkshire Terrier	F	P	2 m	N	N/A
DIAPD-3485/3	2019	Veneto	Bloodhound	M	P	2 m	Y	N/A
DIAPD-50725/5	2019	Veneto	N/A	N/A	P	2 m	Y	N/A
DIAPD-51808/4	2019	Veneto	French Bulldog	M	P	2 m	N/A	N/A
DIAPD-52182/8	2019	Trentino Alto Adige	Cavalier King Charles Spaniel	M	P	2 m	N/A	N/A
DIAPD-55450/9	2019	Veneto	German Shepherd	M	P	4 m	N/A	N/A

* for sample DIAPD-52386/3, the information regarding the type of vaccine used is not available; sample DIAPD-54123/2 was vaccinated with Vaguardi7.

**Table 2 viruses-14-00917-t002:** Point-mutations identified in IZSVe sequences. Point-mutations in the VP2 gene of IZSVe sequences, the type of mutation (silent: sil, missense: mis and non-sense: ns) and the corresponding amino acid change (AA change). -: no changes identified.

Sequence Presenting the Mutation in the VP2 Gene	Phylogenetic Group	VP2 Nucleotidic Residue																									
		156	268	459	633	812	882	923	936	946	1002	1008	1020	1024	1039	1043	1053	1054	1057	1079	1091	1126	1174	1182	1297	1380	1485
MT996031 CPV2-Canis-Italy-TrentinoAltoAdige-strain 18DIAPD-50765-13-2018-new2a	2A.3	G	-	-	-	-	-	-	-	-	-	-	-	-	-	-	-	-	-	-	-	-	A	-	-	-	-
MT996032 CPV2-Canis-Italy-TrentinoAltoAdige-strain 18DIAPD-51301-13-2018-new2a	2A.3	G	-	-	-	-	-	-	-	-	-	-	-	-	-	-	-	-	-	-	-	-	-	-	-	-	-
MT996030 CPV2-Canis-Italy-Puglia-strain DIAPD 18DIAPD-3417-6-2018-new2b	2B	-	C	-	-	-	-	-	-	-	-	-	-	-	-	-	-	-	-	-	-	-	-	-	-	-	-
MT996020 CPV2-Canis-Italy-FriuliVeneziaGiulia-strain 18DIAPD-1100-3-2018-new2b	2B	-	-	C	-	-	-	-	-	-	-	-	-	-	-	-	-	-	-	-	-	-	-	-	-	-	-
MT996025 CPV2-Canis-Italy-FriuliVeneziaGiulia-strain 18DIAPD-53807-2-2018-new2b	2B	-	-	C	-	-	-	-	-	-	-	-	-	-	-	-	-	-	-	-	-	-	-	-	-	-	-
MT996036 CPV2-Canis-Italy-Veneto-strain 17DIAPD-52386-3-2017-new2b	2B	-	-	C	-	-	-	-	-	-	-	-	-	-	-	-	-	-	-	-	-	-	-	-	-	-	-
MT996022 CPV2-Canis-Italy-FriuliVeneziaGiulia-strain 18DIAPD-51020-3-2018-OR	OR	-	-	-	C	G	-	-	-	A	-	-	-	-	-	-	-	-	-	-	-	-	-	-	-	-	-
MT996057 CPV2-Canis-Italy-Veneto-strain 19DIAPD-3485-3-2019-OR	OR	-	-	-	-	G	-	-	-	A	-	-	-	-	-	-	-	-	-	-	-	-	-	-	-	-	-
MT996019 CPV2-Canis-Italy-FriuliVeneziaGiulia-strain 17DIAPD-57861-2-2017-new2a	/	-	-	-	-	-	A	-	-	-	-	-	-	-	-	-	-	-	-	-	-	-	-	-	-	-	-
OM525670 CPV2-Canis-Italy-Veneto-13VIR2697-8-2013-new2b	2B	-	-	-	-	-	-	G	G	-	A	-	T	-	T	T	-	-	-	A	-	-	-	-	-	-	-
MT996047 CPV2-Canis-Italy-Veneto-strain 18DIAPD-55509-2-2018-new2b	2B	-	-	-	-	-	-	-	-	-	A	-	-	-	-	-	-	-	-	-	-	-	-	-	-	-	-
OM525673 CPV2-Canis-Italy-TrentinoAltoAdige-14VIR3493-3-2014-new2b	2B	-	-	-	-	-	-	-	-	-	-	G	-	-	-	-	-	-	-	-	G	-	-	-	-	-	-
MT996023 CPV2-Canis-Italy-FriuliVeneziaGiulia-strain 18DIAPD-52707-2-2018-new2a	/	-	-	-	-	-	-	-	-	-	-	-	-	C	-	-	-	-	-	-	-	-	-	-	-	-	-
MT996044 CPV2-Canis-Italy-Veneto-strain 18DIAPD-3617-4-2018-new2a	2A.3	-	-	-	-	-	-	-	-	-	-	-	-	-	-	-	A	-	-	-	-	-	-	-	-	-	-
MT996058 CPV2-Canis-Italy-Veneto-strain 19DIAPD-50725-5-2019-2c	2C.1	-	-	-	-	-	-	-	-	-	-	-	-	-	-	-	A	-	-	-	-	-	-	T	-	-	-
OM525667 CPV2-Canis-Italy-Veneto-15VIR1684-6-2015-2c	2C.1	-	-	-	-	-	-	-	-	-	-	-	-	-	-	-	-	T	A	-	-	-	-	-	-	-	-
OM525674 CPV2-Canis-Italy-TrentinoAltoAdige-14VIR3058-2-2014-new2a	2A.1	-	-	-	-	-	-	-	-	-	-	-	-	-	-	-	-	-	-	-	-	A	-	-	-	-	-
MT996035 CPV2-Canis-Italy-Veneto-strain 17DIAPD-2046-16-2017-2c	2C.1	-	-	-	-	-	-	-	-	-	-	-	-	-	-	-	-	-	-	-	-	-	-	-	T	-	-
OM525666 CPV2-Canis-Italy-Veneto-15VIR1838-10-2015-new2a	2A.1	-	-	-	-	-	-	-	-	-	-	-	-	-	-	-	-	-	-	-	-	-	-	-	-	C	-
MT996026 CPV2-Canis-Italy-FriuliVeneziaGiulia-strain 18DIAPD-55181-3-2018-new2a	2A.2	-	-	-	-	-	-	-	-	-	-	-	-	-	-	-	-	-	-	-	-	-	-	-	-	-	C
MT996027 CPV2-Canis-Italy-FriuliVeneziaGiulia-strain 18DIAPD-55182-3-2018-new2a	2A.2	-	-	-	-	-	-	-	-	-	-	-	-	-	-	-	-	-	-	-	-	-	-	-	-	-	C
MT996050 CPV2-Canis-Italy-Veneto-strain 18DIAPD-855-18-2018-new2a	2A.2	-	-	-	-	-	-	-	-	-	-	-	-	-	-	-	-	-	-	-	-	-	-	-	-	-	C
Nucleotidic mutation		A156G	A268C	T459C	T633C	A812G	G882A	T923G	A936G	G946A	T1002A	T1008G	A1020T	T1024C	G1039T	C1043T	G1053A	C1054T	T1057A	G1079A	C1091G	C1126A	G1174A	A1182T	A1297T	A1380C	T1485C
Frequency in the Italian population (%)		0.4	0.2	0.6	0.2	0.3	0.2	0.2	0.2	0.3	0.3	0.2	0.2	0.2	0.2	0.2	0.3	0.2	0.2	0.2	0.2	0.2	0.2	0.2	0.2	0.2	0.6
Silent/missense/non sense		sil	mis	sil	sil	mis	sil	mis	sil	mis	sil	sil	sil	mis	mis	mis	sil	mis	mis	mis	mis	mis	mis	sil	mis	sil	sil
AA change			T90P			K271R		V308G		V316I				Y342H	A347S	S348F		P352S	F353I	G360E	A364G	P376T	G392R		T433S		

## Data Availability

The data presented in this study are openly available in GenBank.

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
