# Peer review of "Molecular Investigation of Recent Canine Parvovirus-2 (CPV-2) in Italy Revealed Distinct Clustering"

_viruses, 2022, doi:10.3390/v14050917_

Round 1

Reviewer 1 Report

I suggest authors either to add Discussion section, or to rename Results section into Results and discussion as it is missing, obviously, by technical mistake. 

Author Response

Dear Editor,

We thank the reviewers for the useful comments and suggestions provided in order to render the paper more readable and useful for the scientific community. We have carefully addressed all the comments of both reviewers modifying the paper. We hope this version now meets the quality standards of Viruses.

Responses to reviewer 1 are provided after each comment in italics. Revisions have been highlighted in yellow in the revised version of the manuscript.

Reviewer 1

I suggest authors either to add Discussion section, or to rename Results section into Results and discussion as it is missing, obviously, by technical mistake.

R: we thank the referee and we renamed the result section as “Result and Discussion”

Reviewer 2 Report

This is an interesting study presenting the evolutionary drift of CPV in Northern Italy, collected from specimens of symptomatic dogs during the past years. The analyses are based on a significant number of sequences of the VP2 region and were analyzed and presented according to established standards. In addition, the overall study appears significant and could have potential impact in order to prevent uncontrolled spread and potential occurrence of novel variants. However, a major concern about the presentation of the results consists in the presentation of the Figures, summarizing the most important findings. In particular, Fig. 2 is hardly understandable, since annotations are much too small to read in the printed version and Fig. 2B lacks information, in order to document a potential transfer of CPV variants from Hungary towards Italy. In Addition, Fig. 3 would benefit from additional/clearer information to obtain an overview regarding the new sequences obtained here – again in absence of readability this Figure is not providing appropriate information. Regarding the sequencing results, it would have been beneficial to obtain information regarding the NS-region. This could potentially help to determine, whether some sequences of old(er) CPV-strains derived from vaccine- and/or wild type strains.

Author Response

Dear Editor,

We thank the reviewers for the useful comments and suggestions provided in order to render the paper more readable and useful for the scientific community. We have carefully addressed all the comments of both reviewers modifying the paper. We hope this version now meets the quality standards of Viruses.

Responses to reviewer 2 are provided after each comment in italics. Revisions have been highlighted in yellow in the revised version of the manuscript.

Reviewer 2

This is an interesting study presenting the evolutionary drift of CPV in Northern Italy, collected from specimens of symptomatic dogs during the past years. The analyses are based on a significant number of sequences of the VP2 region and were analyzed and presented according to established standards. In addition, the overall study appears significant and could have potential impact in order to prevent uncontrolled spread and potential occurrence of novel variants. However, a major concern about the presentation of the results consists in the presentation of the Figures, summarizing the most important findings.

R: we thank the referee for its suggestions and as specified below we have modified figures 2 and 3.

In particular, Fig. 2 is hardly understandable, since annotations are much too small to read in the printed version and Fig. 2B lacks information, in order to document a potential transfer of CPV variants from Hungary towards Italy.

R: Figure 2 was modified by delating figure 2b as it does not add fundamental information and decrease the resolution of Figure 2A

In Addition, Fig. 3 would benefit from additional/clearer information to obtain an overview regarding the new sequences obtained here – again in absence of readability this Figure is not providing appropriate information.

R: we thank the referee for this comment and we modified the figure in order to render it more readable by increasing the resolution.

Regarding the sequencing results, it would have been beneficial to obtain information regarding the NS-region. This could potentially help to determine, whether some sequences of old(er) CPV-strains derived from vaccine- and/or wild type strains.

R: regarding the lack of sequencing of the NS gene, we agree with the referee. The reason why such sequences were not generated and therefore not included in the paper is due to the difficulties in using the internal NGS platform as overloaded with the SARS-CoV2 sequences and with the lack of funds to use an external services. However, a second NGS platform will be available soon and CPV-2 NS sequences will be generated together with CPV-2 samples collected during the interregional surveillance program between North East Italy and Austria on illegally imported pet animals. Such limit of the study was reported in the conclusion section.

Round 2

Reviewer 2 Report

I appreciate the author's efforts to improve the manuscript, particularly the visibility of the Figures. I believe these modifications significantly improved the mansucript.